# Quantum Generative Adversarial Networks for High Energy Physics Simulations

Rey Guadarrama, Sergei Glayzer, Konstantin Matchev, Katia Matcheva, Kyoungchul Kong, Gopal Ramesh Dahale, Isabel Pedraza, Haydee Hernandez-Arellano, Mariia Baidachna

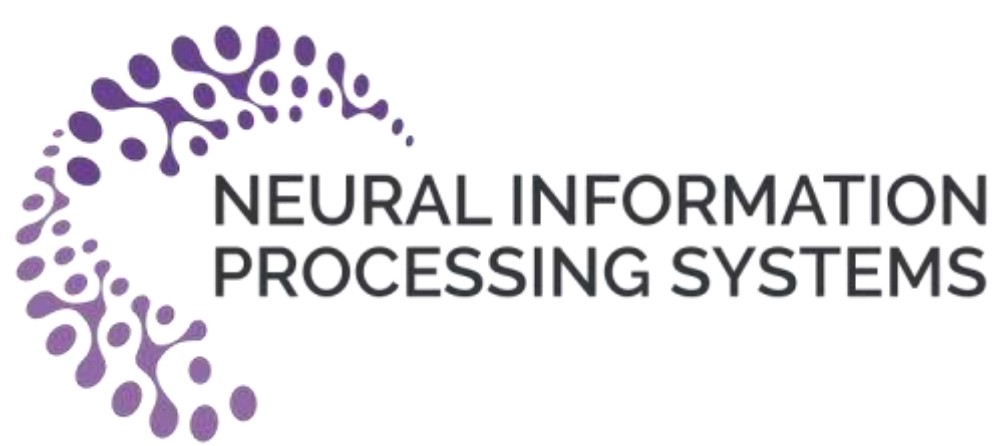

## Introduction

Monte Carlo simulations are essential in computational physics and high-energy physics (HEP) for modeling complex systems and particle interactions, but face scalability challenges due to their computational intensity. Generative Adversarial Networks (GANs) present a promising alternative for generating realistic data [1] but face challenges such as vanishing gradients, mode collapse, and significant computational requirements. Recent studies highlight quantum algorithms, particularly Quantum Generative Adversarial Networks (qGANs), as potential solutions. Zoufal et al [2]. demonstrated that qGANs can efficiently learn and encode probability distributions into quantum states, making them a compelling approach for future HEP simulations. This work presents the implementation of a qGAN to generate gluon-initiated jet images from ECAL detector data, a task crucial for high-energy physics simulations at the Large Hadron Collider (LHC). The results demonstrate high fidelity in replicating energy deposit patterns and preserving the implicit training data features

## Methodology

We used Andrews et al.'s [3] dataset from QCD dijet simulations on the CERN CMS Open Data Portal, containing 933,206 three-channel, 125×125 images of quarks and gluons. Only the ECAL channel was used. Gluon images were cropped to 80×80 and downscaled via sum pooling to 8×8 to represent scaled ECAL energy deposits. Our method uses Huang et al.'s [4] hybrid architecture with quantum generators and a classical discriminator. Quantum generators construct images from quantum basis state probabilities along with a post-processing step to get rid of the normalization constraints, representing pixel intensities. They employ PQCs initialized with a uniform reference state $|\Psi_{in}\rangle$, parameterized Pauli-Y rotations, CZ gates, and an auxiliary register for a non-linear transformation (figure 1). The classical discriminator is a dense neural network with input (64 nodes), hidden (128 and 32 nodes, ReLU), and Sigmoid output layers.

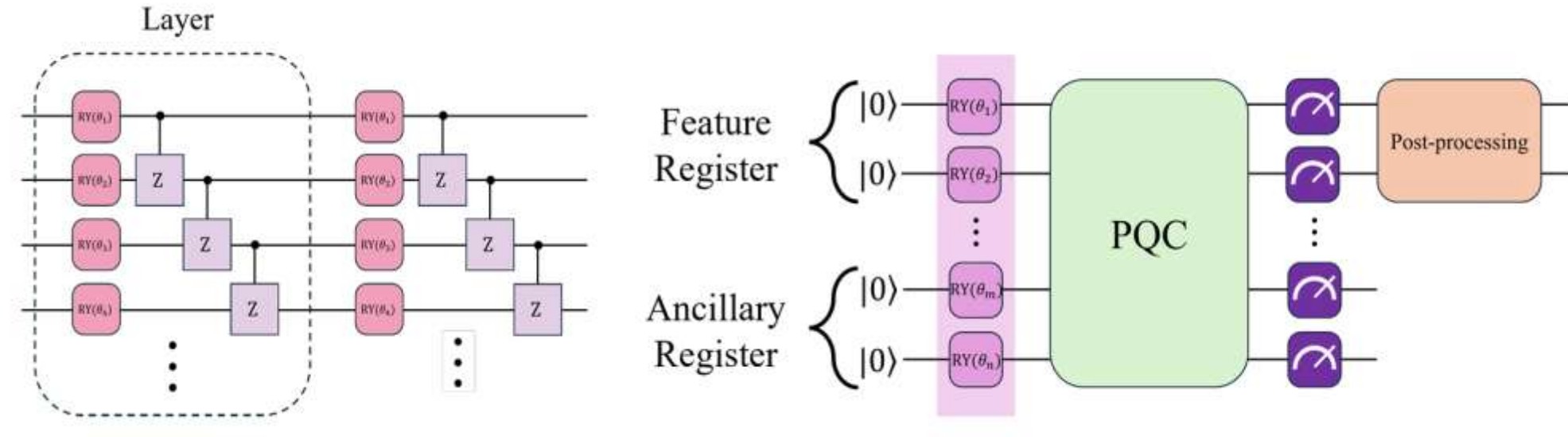

**Figure 1**: Schematic representation of the quantum generator. A single layer (left) consisting of Pauli-Y rotations followed by a control Z gate entangling block. The PQC (right) includes both a feature register and an ancillary register along with a post-processing step after measurement.

## Results

We train 2 PQCs (5 feature qubits, 2 auxiliary qubits, 5 layers) on 512 gluon images for 30 epochs with a batch size of 1. The evaluation of the generative model shows the evolution of Fréchet Inception Distance (FID) and Root Mean Squared Error (RMSE) in Figure 2, demonstrating fast convergence and increased accuracy in generating realistic gluon images. Figure 3 compares the scaled total energy and rechit energy deposits from real and generated gluon jets, showing high accuracy in reproducing the implicit features of the training data.

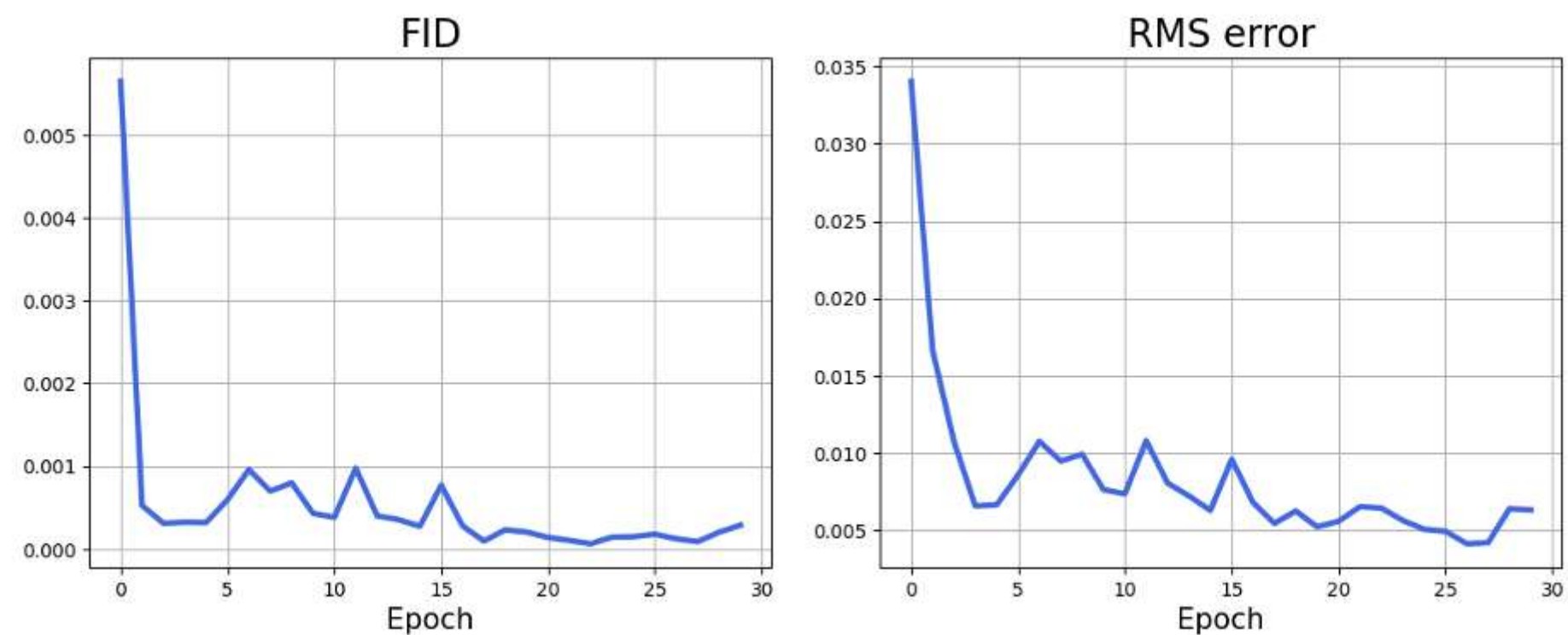

**Figure 2**: Frenchet inception distance (left) and the root mean squared error (right) computed using the training data overlay and the generated data overlay at the end of each epoch.

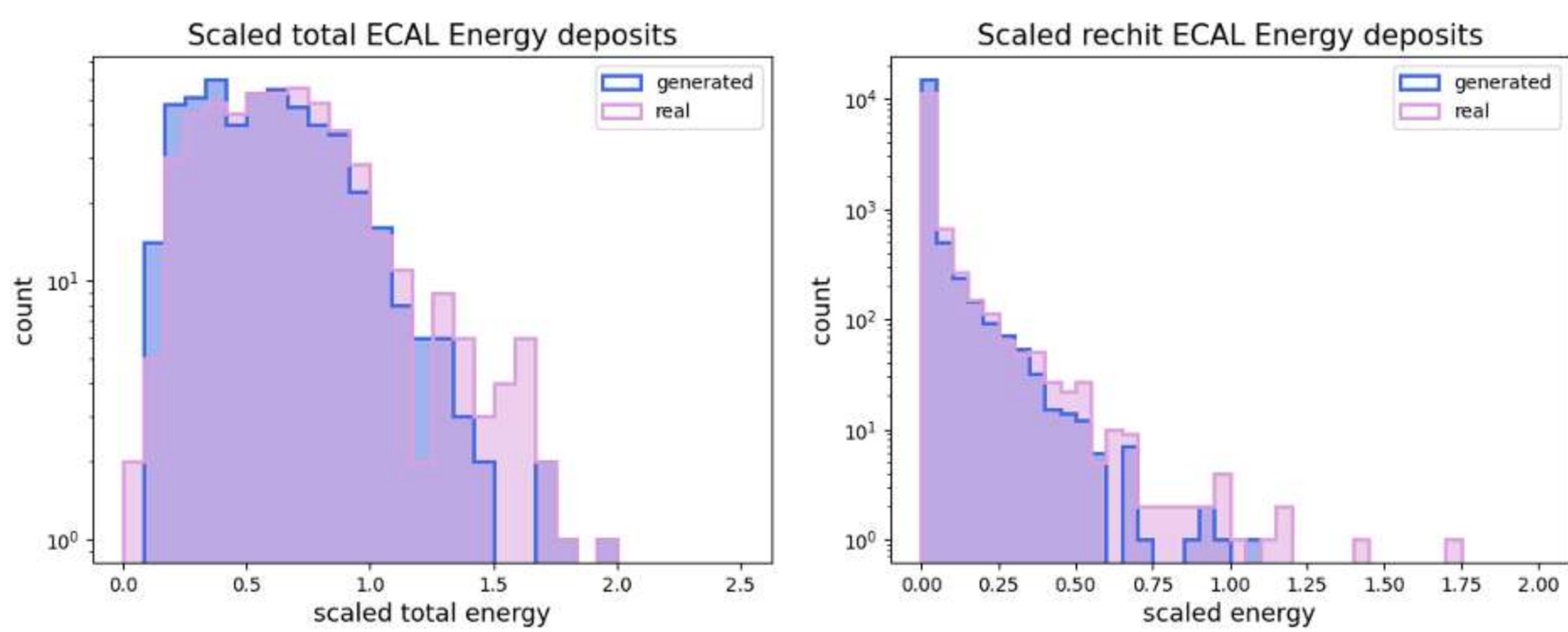

**Figure 3**: Comparison of the generated and real ECAL energy deposits. The left plot shows the scaled total ECAL energy deposits. The right plot depicts the scaled rechit energy deposits. Both plotted in a log scale.

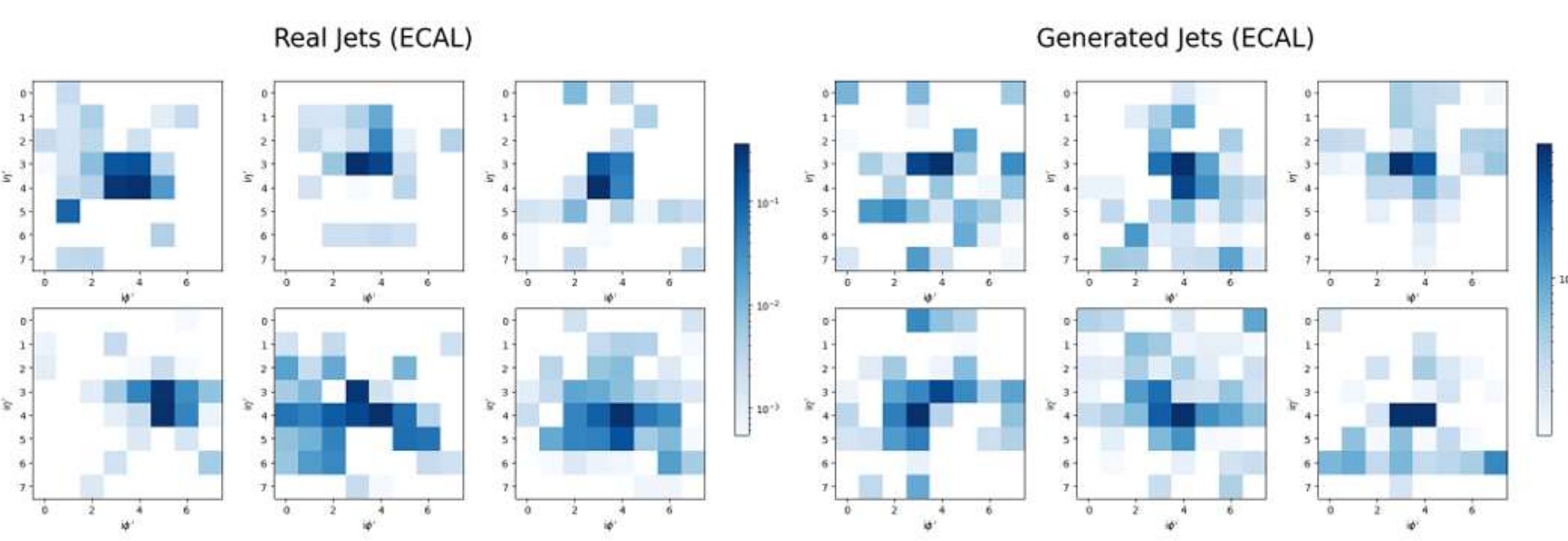

**Figure 4**: Comparison of real (left) and generated (right) ECAL jet images. The generated jet images are generated by the trained model. Both sets are plotted in a log scale.

Figure 4 compares real and generated ECAL jet images, showing similar energy deposition patterns in both, with generated jets closely matching the real ones. Figure 5 compares real and generated ECAL energy deposit overlays, with both showing similar distributions, particularly in the core region, highlighting the qGAN's ability to replicate energy deposit patterns accurately.

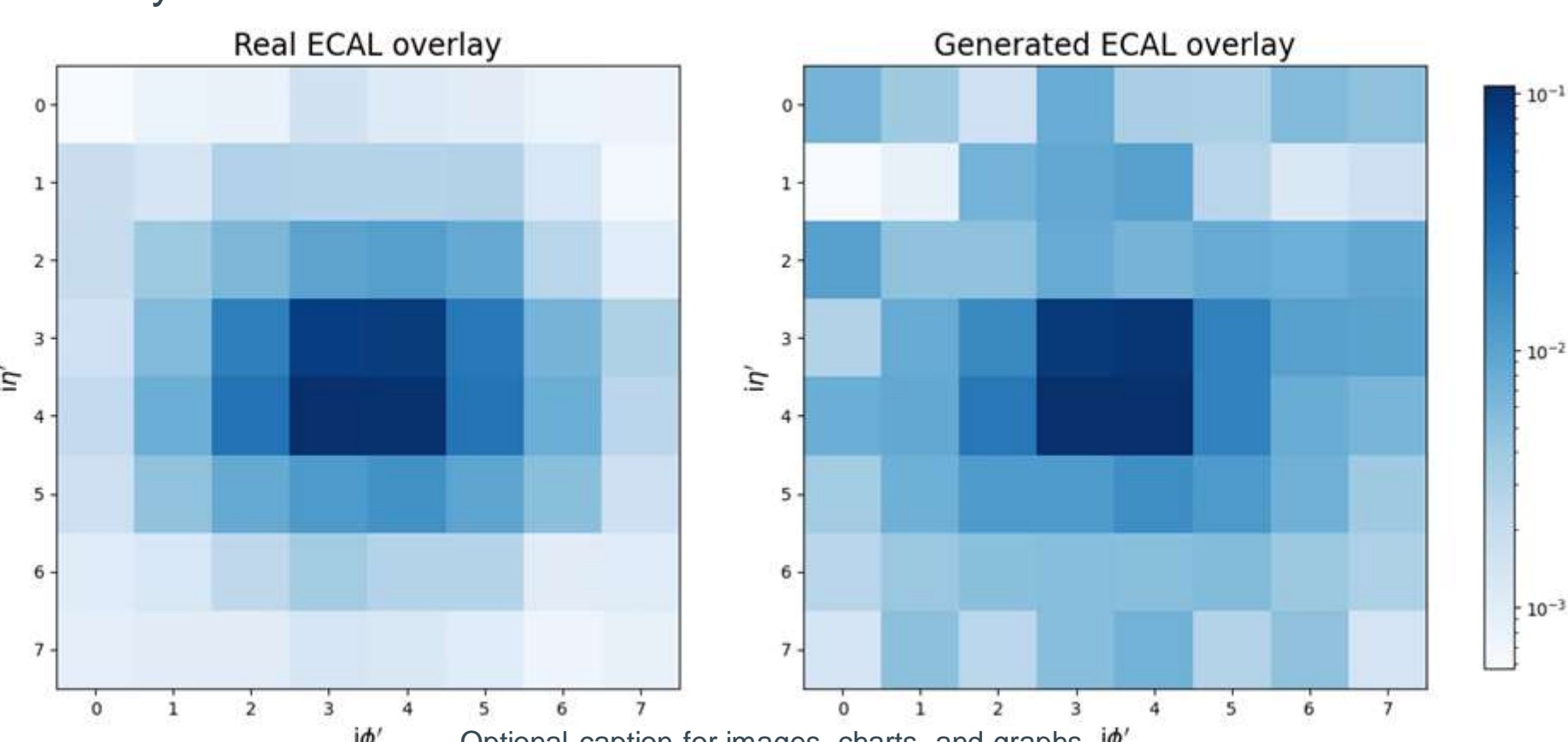

**Figure 5**: Real and generated ECAL jet-view image overlays. The left panel shows the real ECAL overlay, while the right panel shows the generated ECAL overlay over 512 jets each, both in log-scale. Image resolution 8 × 8.

## Conclusions

This work successfully implemented a qGAN to generate ECAL gluon-initiated jet images from CMS Open Data, demonstrating its ability to replicate energy deposit distributions accurately. However, the study was limited to 512 pictures and the ECAL channel, and training was performed on quantum simulators, which may not reflect actual quantum hardware performance. Future work should explore real quantum hardware, expand the dataset, and integrate data from additional sub-detectors to improve scalability and generalization. This study sets the first step for future work on the qGAN model to simultaneously generate three-channel images and incorporate quark jet image generation, improving the simulation capabilities for LHC-related tasks.

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
