# OpenReview forum: "Quantum Generative Adversarial Networks for High Energy Physics Simulations"
_NeurIPS.cc/2024/Workshop/MLNCP — MLNCP Poster_

### Official Review · Reviewer_G5Mo · 2024-09-21
**Recommend more discussion on training scaling**

**Rating:** 6
**Confidence:** 5

**Review:**

The authors consider a hybrid quantum-classical generative-adversarial network (GAN) for producing jet images. In such a setting, the generator of the GAN is a parameterized quantum circuit; an adversarial, classical discriminator network then attempts to distinguish the generated images from true images. Given quantum networks are able to sample efficiently from distributions known to be difficult for classical models to sample from---and as the data set was generating by a quantum process---such a use-case is promising for quantum generative models.

My only concern is on the scaling of training such a network. The authors consider a parameterized quantum circuit, which are known to have bad [local minima](https://www.nature.com/articles/s41467-022-35364-5) or [bad gradient scaling](https://www.nature.com/articles/s41467-021-21728-w). Given the efficiency of training the network is a requirement for the quantum model to be practical, discussion on this point---and proposals for avoiding such issues in training, perhaps by using some kind of network structure suggested by the data set---would be required to bump up my recommendation to a strong accept. Never-the-less, as-is I recommend a weak accept.

---

### Official Review · Reviewer_PWro · 2024-10-02
**qGAN for High Energy Physics Simulations**

**Rating:** 4
**Confidence:** 3

**Review:**

The authors propose how the qGAN model can be applied to generate ECAL detector images.

While the paper was really interesting, the justification for the quantum generator related to this data was not absolutely clear. is it only due for the hypothetical speed-up ? It would have been instructive to at minimum compare results to a deterministic generator (or other probabilistic modelling approaches) to understand if the different means of modelling the data probability brought something to this practial application or not.

Seeing as a quantum simulator was used, and there is no innovation related to how the qGAN approach might be optimsed for a given quantum compute paradigm I question whether this paper is a good fit for this workshop.

---

### Decision · Program_Chairs · 2024-10-10

Accept (Poster)